# Analysis of an Impact of Inertia Parameter in Active Disturbance Rejection Control Structures

**Dariusz Pazderski \***, **Radosław Patelski**, **Bartłomiej Krysiak** and **Krzysztof Kozłowski**

Institute of Automatic Control and Robotics, Poznan University of Technology,
ul. Piotrowo 3a, 60-965 Poznan, Poland; radoslaw.patelski@put.poznan.pl (R.P.);
bartlomiej.krysiak@put.poznan.pl (B.K.); krzysztof.kozlowski@put.poznan.pl (K.K.)
\* Correspondence: dariusz.pazderski@put.poznan.pl

**Abstract:** This paper is focused on a distributed control of fully actuated manipulators under operating conditions when dynamic couplings between their joints are insignificant. The main research aspect was to examine the influence of the inertia parameter on the tracking quality for control systems based on the active rejection paradigm. The theoretical description and preliminary hypothesis were supported by extensive simulation and experimental results. In particular, it is demonstrated that choosing the inertia parameter according to the real dynamics properties does not guarantee the best performance of the considered control structures.

**Keywords:** trajectory tracking control; perturbed systems; active disturbance rejection

---

## 1. Introduction

In the robotics literature, one of important issues is the tracking control of mechanical systems [1]. In the case of model-based control strategies, it is necessary to precisely understand the dynamic model of the plant taking into account its structure and parameterization. In addition, position and velocity (even acceleration) measurements should be available. These requirements are not usually met in practice. Parameters may change over time and state, and methods of obtaining them are time-consuming or computationally expensive. In particular, modeling highly nonlinear effects such as friction forces in mechanical systems can be challenging. Furthermore, measurements are often limited to the known positions, whereas velocities and accelerations are not measured directly.

In general, the model-based control approach has to deal with uncertainties both internal (parametric) and external (disturbances) to the system. To attenuate the undesirable effects induced by internal model uncertainties robust and adaptive control strategies can be used. It is also possible to employ the free model control approach, where disturbance terms are being identified on-line [2]. A similar concept is proposed by the active disturbance rejection approach (ADRC) originally introduced in [3] and later covered in detail in several works including [4–6]. The ADRC methodology is based on a compensation loop that provides the rejection of unmodeled disturbances in the control input path. This leads to the feedback linearized system in the form of integrator chain of the $n$-th order, where $n$ is the relative degree of the system. Thus, use of a direct and active estimation and rejection transforms the system to a linear one, for which a solution to a simple control problem of the outer loop design can be proposed. Very often, it is solved by the state feedback and feed-forward controller. This is a well known method, discussed in [7]. ADRC has been successfully used for applications which require robust performance, including control of brushless DC motor with unknown friction model [8] or gasoline engines with variable valve timings [9]. Disturbance rejection method is often applied in combination with other control paradigms, for example by using a generalized proportional integral (GPI) observer for a wire haptic systems [10] or PMSM servo [11]. Another solution employing ADRC with predictive functional controller (PFC) has

been developed for the latter in [12]. Some insights on practical aspects of ADRC implementation in an industrial environment has also been presented in [13,14].

The tracking control task along with ADRC is often supported by the assumption that the time derivative of the reference input and the feed-forward signal have to be computed. However, some research presents the possibility of weakening these requirements [15–17] by the application of ADRC in the closed-loop (this structure is known as EADRC). The important issues concerning the utilization of ADRC are stability properties [18–20] with respect to different kinds of disturbances, including discontinuous disturbances and the incorrect estimation of the system degree [21] or input gain [22].

This study investigated the impact of the selection of the inertia parameter on the closed-loop system performance, especially the tracking quality, where the tracking controller designed for a mechanical system is based on the active rejection paradigm. Some research has already been carried out to find a range of this parameter for which a closed-loop system remains stable [23–25]. In recent work [26], an analytical solution to this problem has been investigated for a second order system. This result indicates that the range of acceptable parameter values is relatively large. However, a relevant question concerning the influence of the inertia parameter selection within this range on the control performance, seems to be omitted in previous research. One of the aim of this paper is to fulfill this gap. To solve this problem we take into account two control structures, in which the observer is designed for the process model (ADRC) or for the closed-loop system (EADRC).

Apart from simulation studies, we carried out extensive experimental work. For this purpose we use a robotic astronomical mount equipped with 0.5 m class telescope [27]. Certain studies concerning its mechanical properties in terms of tracking control have been reported in [28].

The rest of the paper is organized as follows. Section 2 discuses the controller design, along with the model of the process, the controller based on ADRC and EADRC, and the comparison between of those two control structures. Section 3 presents the simulation results, whereas in Section 4, the experiments using the astronomical mount are given. Finally, Section 5 concludes the paper.

## 2. Control Design

### 2.1. Model

Consider the following fully actuated second order mechanical system:

$$M(q)\ddot{q} + C(q,\dot{q})\dot{q} + G(q) + F(\dot{q}) = \tau, \tag{1}$$

where $q = [q_1 \ q_2 \ \ldots \ q_n]^\top \in \mathbb{R}^n$ denotes the configuration, $M(q) \in \mathbb{R}^{n \times n}$ is the inertia matrix, $C(q,\dot{q}) \in \mathbb{R}^{n \times n}$ stands for Coriolis and centrifugal terms, $F(\dot{q}) \in \mathbb{R}^n$ describes friction effects, $G(q) \in \mathbb{R}^n$ defines gravity forces, and $\tau = [\tau_1 \ \tau_2 \ \ldots \ \tau_n]^\top \in \mathbb{R}^n$ is an input. Computing acceleration $\ddot{q}$ one obtains:

$$\ddot{q} = M^{-1}(q)\left(-C(q,\dot{q})\dot{q} - G(q) - F(\dot{q}) + \tau\right). \tag{2}$$

Equivalently, one can decompose the system (2) into $n$ smaller subsystems, cf. Figure 1, which satisfy the following scalar equation

$$\ddot{q}_i = m_{ii}^{-1}(q)\tau_i + h_i(q,\dot{q},\tau), \tag{3}$$

where $i = 1, \ldots, n$, $m_{ii}^{-1}(q)$ stands for the entry $(i,i)$ of the mass matrix inverse $M^{-1}(q)$ and

$$h_i(q,\dot{q},\tau) = \underline{m}_i(q)\left(-C(q,\dot{q})\dot{q} - G(q) - F(\dot{q}) + W_i\tau\right), \tag{4}$$

where $\underline{m}_i$ is the $i$th row of $M^{-1}(q)$, $W_i \in \mathbb{R}^{n \times n}$ is defined from the identity matrix by zeroing its entry $(i,i)$.

To facilitate the control design, we define the state of subsystem (3) by $x = [x_1 \ x_2]^\top := [q_i \ \dot{q}_i]^\top$ and rewrite dynamics (3) as follows

$$\dot{x} = \begin{bmatrix} x_2 \\ \frac{1}{J(x_1, q^*)} u + h(x_1, x_2, q^*, \dot{q}^*, \tau) \end{bmatrix}, \qquad (5)$$

where $q^*$ and $\dot{q}^*$ are entries of configuration $q$ and its velocity $\dot{q}$ other than $q_i$ and $\dot{q}_i$, respectively. The term $u$ stands for $\tau_i$, $J(x_1, q^*) := m_{ii}(q)$, whereas $h(x_1, x_2, q^*, \dot{q}^*, \tau) = h_i(q, \dot{q}, \tau)$ is an aggregated component of the dynamics.

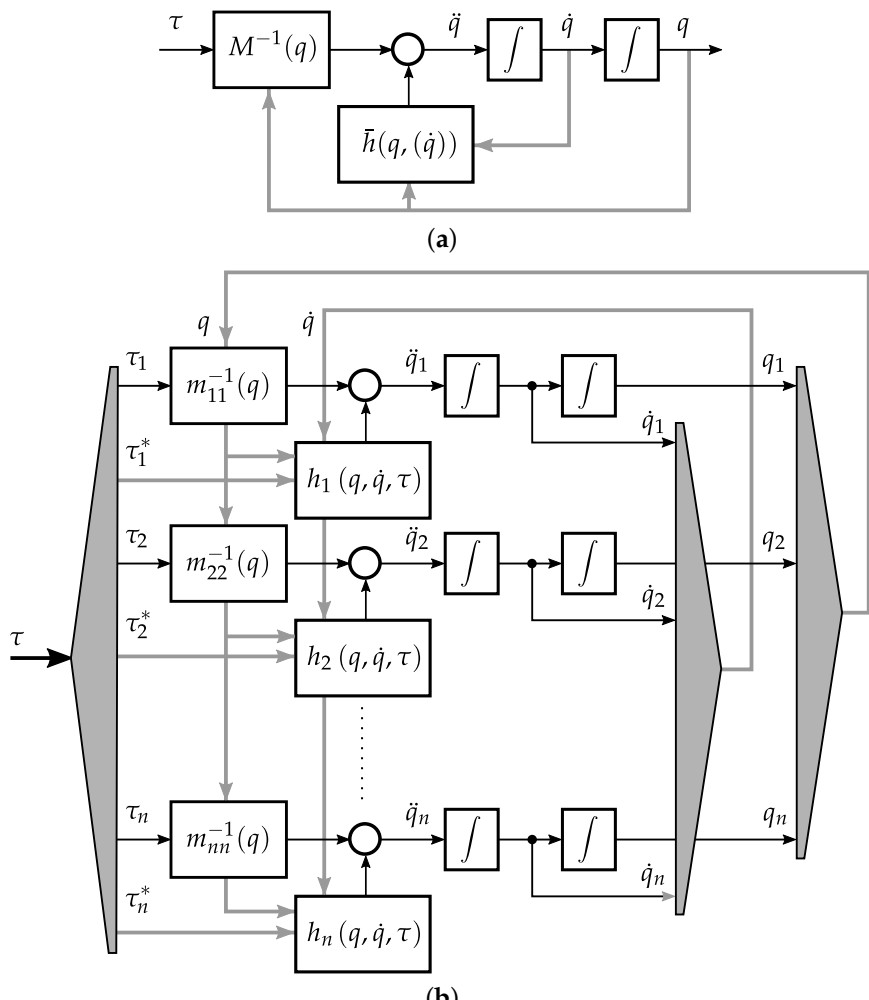

**Figure 1.** Block diagrams of the considered fully actuated mechanical system: (**a**) centralized model (2) with $\bar{h} := M^{-1}(q) \left(-C(q, \dot{q})\dot{q} - G(q) - F(\dot{q})\right)$, (**b**) distributed model which consists of $n$ subsystems defined by (3) where $\tau_i^* := W_i \tau$.

## 2.2. Design of a Nominal Tracking Controller

Let $r(t) \in \mathbb{R}$ be a time dependent function of a class $C^3$ which prescribes a reference trajectory. To quantify the tracking error in terms of position and velocity we define

$$e = [e_1 \ e_2]^\top := x - x_r, \qquad (6)$$

where $x_r = [r\ \dot{r}]^\top \in \mathbb{R}^2$ is the reference state. Consequently, taking into account system (5) one obtains the following open-loop error dynamics

$$\dot{e} = \begin{bmatrix} e_2 \\ \frac{1}{J(x_1,q^*)} u + h(x_1, x_2, q^*, \dot{q}^*, \tau) - \ddot{r} \end{bmatrix}.$$ (7)

In order to stabilize system (7) at $e = 0$ one could apply the following feedback

$$u := J(x_1, q^*)\left(-Ke - h(x_1, x_2, q^*, \dot{q}^*, \tau) + \ddot{r}\right),$$ (8)

where $K = [k_p\ k_d]$ is the gain matrix with $k_p, k_d > 0$. However, the application of controller (8) may be inconvenient as a result of an imprecise knowledge of the terms $J, h$, along with the unavailability of velocity measurements. In addition, the control law (8) is not completely decentralized, since the information about states and controls of each subsystem is required, cf. the terms $q^*, \dot{q}^*$, and $\tau$.

To cope with these issues, we considered a modified version of the control law (8) and assumed that the controller input is described by

$$u := \hat{J}\left(\gamma u_f - u_c\right),$$ (9)

where $u_f$ defines a stabilizing feedback and $u_c$ is a function partly responsible for at least partial rejection of terms in the input path. The term $\hat{J}$ stands for the designed inertia parameter and $\gamma > 0$ is a positive constant that additionally enables scaling of the feedback. The rejection term $u_c$ can be estimated by a high-gain extended state observer (HG ESO). In this study, we employed the active disturbance rejection paradigm (ADR) and considered two control structures, which are described in a sequel.

### 2.3. ESO Design Based on the Process Dynamics (ADRC)

Here, we recall a classic structure of ESO, which is based on the original process dynamics (5). We treat the term $h$ in (5) as an unknown disturbance and consider the following estimation law

$$\dot{\hat{z}} = \begin{bmatrix} \hat{z}_2 \\ \hat{z}_3 + \hat{J}^{-1} u \\ 0 \end{bmatrix} + L\left(x_1 - \hat{z}_1\right),$$ (10)

where $\hat{z} = [\hat{z}_1\ \hat{z}_2\ \hat{z}_3]^\top \in \mathbb{R}^3$ denotes the estimate, $L \in \mathbb{R}^3$ is the observer gain (chosen in order to guarantee a stable estimation process). Taking into account Formula (9) and employing the classic ADRC approach, we have

$$u_c := \hat{z}_3 - \ddot{r}.$$ (11)

Consequently, Formula (10) can be rewritten as

$$\dot{\hat{z}} = \begin{bmatrix} \hat{z}_2 \\ \gamma u_f + \ddot{r} \\ 0 \end{bmatrix} + L\left(x_1 - \hat{z}_1\right).$$ (12)

Next, we augment the state of system (5) and define $z := \begin{bmatrix} x^\top z_3 \end{bmatrix}^\top$, where $z_3 := \alpha^{-1} h(x_1, x_2, q^*, \dot{q}^*, \tau)$ is a new state with

$$\alpha := \hat{J} J^{-1}(x_1, q^*) > 0$$ (13)

being a dimensionless scaling parameter, representing error of the inertia parameter estimation. Clearly, choosing the inertia estimate $\hat{J}$ greater than the real value of the inertia $J$ leads to the increase of $\alpha$ parameter. In the nominal case, when inertia of the plant is perfectly known and $\hat{J} = J$, $\alpha = 1$.

Applying Formula (9) with (11) in (5), taking into account (13) and definition of $z$, we obtain the following augmented dynamics

$$\dot{z} = \begin{bmatrix} x_2 \\ \alpha z_3 + \gamma \alpha u_f - \alpha \hat{z}_3 + \alpha \ddot{r} \\ \sigma \end{bmatrix}, \tag{14}$$

where $\sigma := \frac{d}{dt}\left\{\alpha^{-1}h(x_1, x_2, q^*, \dot{q}^*, \tau)\right\}$ is time derivative of the total disturbance of the system, which consists of all dynamics that are not explicitly modeled in a controller synthesis. One can notice that $\sigma = \dot{z}_3$. We assume here that $\sigma$ is at least bounded in finite time intervals.

Considering the estimation error $\tilde{z} := \hat{z} - z$, taking time derivative of $\tilde{z}$, using (14) and (10) one obtains

$$\dot{\tilde{z}} = \begin{bmatrix} \tilde{z}_2 \\ \alpha \tilde{z}_3 + \gamma\left(1 - \alpha\right)u_f + \left(1 - \alpha\right)\ddot{r} \\ -\sigma \end{bmatrix} - L\tilde{z}_1. \tag{15}$$

Obviously, the stability of (15) can be achieved only when the terms $u_f$, $\ddot{r}$ and $\sigma$ are prescribed explicitly. However, for a typical mechanical system a sufficient stability requirement is to use high gain coefficients in $L$ that can be selected based on a simple pole placement strategy.

### 2.4. ESO Design Based on the Tracking Error Dynamics (EADRC)

Alternatively, ESO observer can be designed for the tracking error dynamics (7). Assuming that $e_1$ is the measured output, the estimation law (10) is redefined as

$$\dot{\hat{z}} = \begin{bmatrix} \hat{z}_2 \\ \hat{z}_3 + \hat{J}^{-1}u \\ 0 \end{bmatrix} + L\left(e_1 - \hat{z}_1\right). \tag{16}$$

Next, using EADRC strategy and recalling (9) we assume that

$$u_c := \hat{z}_3. \tag{17}$$

Hence, we have

$$\dot{\hat{z}} = \begin{bmatrix} \hat{z}_2 \\ \gamma u_f \\ 0 \end{bmatrix} + L\left(e_1 - \hat{z}_1\right). \tag{18}$$

Now, the extended state is represented by $z := \begin{bmatrix} e^\top z_3 \end{bmatrix}^\top$, where $z_3 := \alpha^{-1}(h(x_1, x_2, q^*, \dot{q}^*, \tau) - \ddot{r}) = \alpha^{-1}(h(e_1 + r, e_2 + \dot{r}, q^*, \dot{q}^*, \tau) - \ddot{r})$.

Then the augmented dynamics of (7) takes the form of

$$\dot{z} = \begin{bmatrix} e_2 \\ \alpha z_3 + \gamma \alpha u_f - \alpha \hat{z}_3 \\ \sigma \end{bmatrix}, \tag{19}$$

where $\sigma = \frac{d}{dt}\left\{\alpha^{-1}(h(e_1 + r, e_2 + \dot{r}, q^*, \dot{q}^*, \tau) - \ddot{r})\right\}$. As in the previous case, we consider the following estimation error dynamics

$$\dot{\tilde{z}} = \begin{bmatrix} \tilde{z}_2 \\ \alpha \tilde{z}_3 + \gamma\left(1 - \alpha\right)u_f \\ -\sigma \end{bmatrix} - L\tilde{z}_1, \tag{20}$$

that differs from (15) by the lack of an explicit presence of the scaled trajectory reference in the derivative of $\tilde{z}_2$. The significance of this difference is shown in subsequent sections.

### 2.5. A Comparison of Two Control Structures

Two control structures are illustrated in Figure 2. For the sake of simplicity, we assume that $\sigma = 0$ and the gain $L$ is chosen as follows: $L = L(\omega_o) := \begin{bmatrix} 3\omega_o & 3\omega_o^2 & \omega_o^3 \end{bmatrix}^\top$, where $\omega_o > 0$ determines the observer bandwidth. In addition, we take into account a perfect tracking condition that is characterized by a zero feedback signal $u_f = 0$. The aim of this analysis was to find how the selection of parameter $\alpha$ affects the steady state estimation errors.

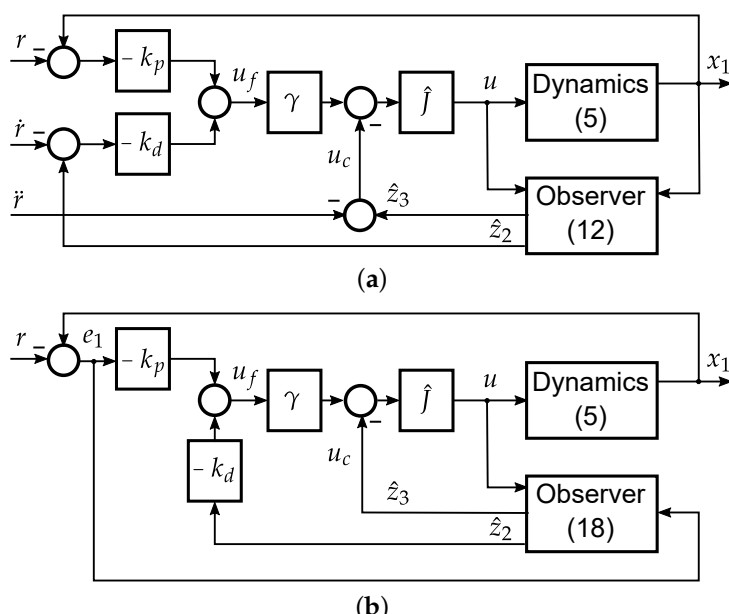

**(a)**

**(b)**

**Figure 2.** Block diagrams of the control structures (in the parenthesis essential mathematical formulas are recalled): (**a**) ADRC, (**b**) EADRC.

Taking above conditions into account dynamics (15) can be written as

$$\dot{\tilde{z}} = \begin{bmatrix} \tilde{z}_2 \\ \alpha\tilde{z}_3 + (1-\alpha)\,\ddot{r} \\ 0 \end{bmatrix} - L\tilde{z}_1. \tag{21}$$

Then it can be proved that for $\dddot{r} = 0$ there exists an equilibrium for system (21) defined at

$$\tilde{z}^* = \begin{bmatrix} 0 & 0 & \left(1-\alpha^{-1}\right)\ddot{r} \end{bmatrix}^\top. \tag{22}$$

Recalling (14) one can write the following:

$$\dot{\tilde{z}}_2 = -\alpha\tilde{z}_3 + \gamma\alpha u_f + \alpha\ddot{r}. \tag{23}$$

Hence, using (22) and substituting $u_f = 0$ in (23), one has $\dot{\tilde{z}}_2 = \ddot{r}$, which indicates that the acceleration $z_2$ is the same as the reference value $\ddot{r}$ even for $\alpha \neq 1$. Similarly, in the case of the EADRC structure, one can easily show that for $\sigma = 0$ the equilibrium point of system (20) is independent from the selection of $\alpha$ and is determined at the origin.

**Corollary 1.** *The considered computations suggest that for the assumed value of parameter $\hat{J}$, the feed-forward term related to the reference trajectory should be designed specifically for the given inertia $\hat{J}$ instead of the real*

*inertia J. This is due to the fact that the considered control approach allows one to virtually modify the inertia of the mechanical system, according to the chosen parameter $\hat{J}$ that is used for the estimation and the rejection of disturbances.*

An important difference between ADRC and EADRC comes from the way that the feed-forward term is taken into account. Clearly, in the EADRC structure the feed-forward term is fully estimated. Conversely, in the ADRC the feed-forward term is provided directly. However, still a part of this term is estimated when no real inertia parameter $\hat{J}$ is selected. We discuss these issues based on simulation and experimental results.

## 3. Simulations

We compare two control schemes introduced in Sections 2.3 and 2.4. The purpose of this study was to find how the selection of inertia parameter $\hat{J}$ affects the tracking precision.

For simulation purposes we considered the following benchmark system being the simplified version of dynamics (5)

$$J(x_1, q^*) := 1, \ h(x_1, x_2, q^*, \dot{q}^*, \tau) := \mu \left( x_2 + \text{sgn} x_2 \right), \tag{24}$$

where $\mu > 0$. Namely, we assume that the predominant force comes from friction (including viscous and Coulomb terms), while dynamic effects due to dynamic interactions and the gravity can be neglected. Such a dynamic model reflects properties of some class of robotic systems which operates in a range of relatively small velocities. For example, it can be seen as a basic model of one axis of the telescope mount that is used for experimental research described in Section 4.

In the simulations, we took into an account the following four cases:

- CO: classic ADRC (cf. Section 2.3) where only the inertia for the observer design is being changed using parameter $\alpha$, while the inertia parameter for the feedback design is fixed such that $\gamma\alpha = 1$,
- CFO: classic ADRC (cf. Section 2.3) where the inertia parameter for the observer and the feedback design is being changed in the same way with $\gamma = 1$,
- EO: EADRC control structure (cf. Section 2.4), where only the inertia parameter for the observer design is being changed using parameter $\alpha$, while the inertia parameter for the feedback design is fixed such that $\gamma\alpha = 1$,
- EFO: EADRC control structure (cf. Section 2.4) where the inertia parameter for the observer and the feedback design being changed in the same way with $\gamma = 1$.

The feedback gains are chosen as $k_p = 1$ and $k_d = 2$, while the observer gains are computed for $\omega_o = 100$. The reference input trajectory is defined as $r = \sin 2t$. For each control structure we considered a series of simulations conducted for different values of $\mu$. Then for each $\mu \in \{0, 1, 2, 5\}$ we performed 50 simulations for the chosen values of the inertia parameter chosen within the range $\alpha \in [0.1, 5]$ (the feedback term $\gamma$ was adjusted appropriately). Based on the closed-loop system trajectories recorded in the time interval from $t_0 = 30$ s to $t_1 = 40$ s we computed the mean absolute error (MAE) quality factor defined as MAE $= \frac{1}{t_1 - t_0} \int_{t_0}^{t_1} e_1(t)dt$.

The results of the simulations are collected in Figure 3. It is evident that an asymptotic stabilization can be ensured only for the classic ADRC structure in the trivial case when $\mu = 0$ (no friction is present) and $\alpha = 1$ (the perfect knowledge of the inertia is assumed). Conversely, for the EADRC structure the asymptotic stabilization is not achieved since the reference term $\ddot{r}$ has to be estimated by the observer. Taking into account that in the considered case $\ddot{r} \neq 0$, such an estimation can be only approximate.

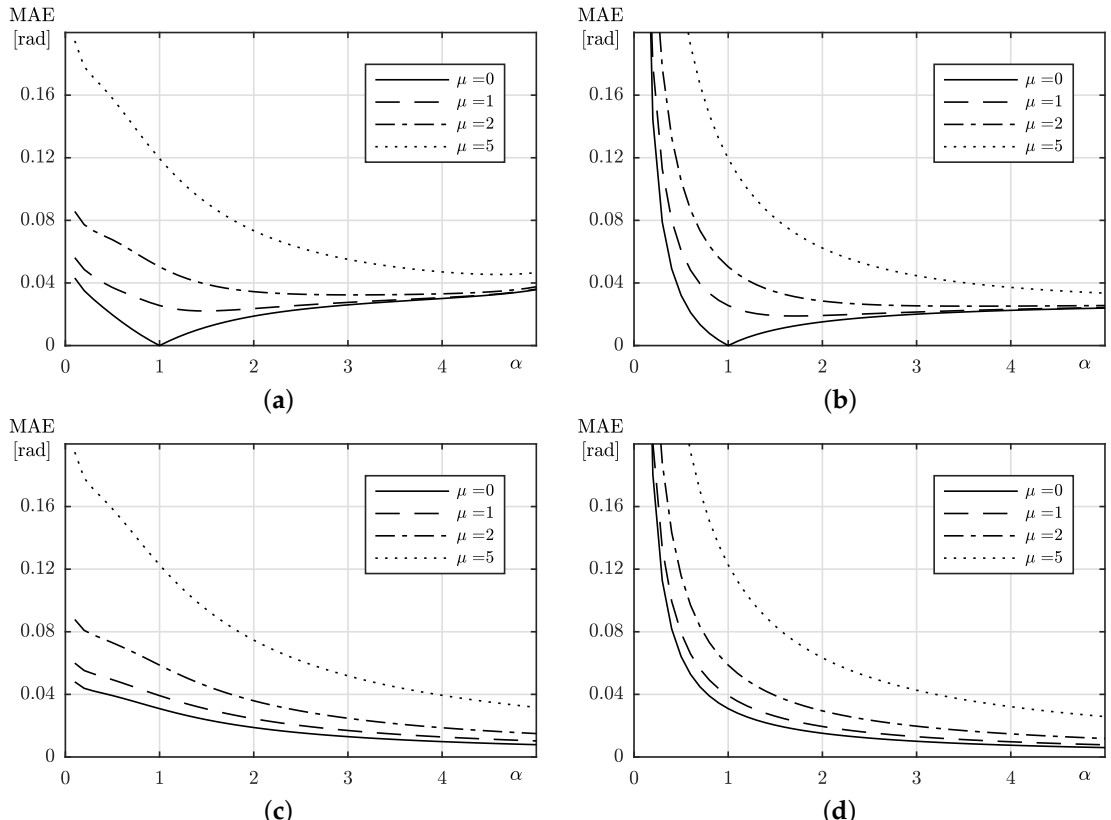

**Figure 3.** Simulation results—mean absolute error (MAE) values obtained for various scenarios: (**a**) CO, (**b**) CFO, (**c**) EO, (**d**) EFO.

An important observation is that increasing of parameter $\hat{J}$ makes it possible to improve the tracking accuracy, especially for the EADRC control structure. For the ADRC structure, the MAE may achieve a local minimum for $\hat{J} \approx J$ when the disturbance is small enough. Then, the terms $u_f$ and $\ddot{r}$ can be perfectly compensated, cf. (15). Despite that, for a higher disturbance, no local minimum was observed.

## 4. Experiments

The experimental research was carried out using the robotized telescope mount, [27], built in the Institute of Automatic Control and Robotics, cf. Figure 4.

The tracking controller was tested for the vertical axis of the mount, whereas the horizontal axis was actively stabilized in the arbitrary chosen fixed position. On the basis of the available dynamic model of the mount, we stated that the inertia momentum of the vertical axis is $J = 35.46 \, \text{kg} \cdot \text{m}^2$. The control parameters were set as follows: feedback gains $k_p = 64$ and $k_d = 16$, observer bandwidth $\omega_o = 200$. We employed a sine reference trajectory $r(t)$ with a frequency $f = 1/12 \, \text{Hz}$, while velocity amplitude $\dot{r}$ was selected as:

The quality of the tracking was compared for ADRC and EADRC controllers depending on the assumed value of the inertia parameter $\hat{J}$. This study was carried out analogously to the study presented in Section 3, i.e., for a control system in which the adopted inertia value was changed in each of the controller parts (i.e., observer, predictive component, and feedback) and for a control system where an ideal knowledge of inertia in the feedback loop was assumed and the estimated inertia was entered only to the observer loop and the feed-forward term.

- $v_{max} = 2\Omega$ (where $\Omega := 7.29 \cdot 10^{-5} \, \text{rad/s}$ is an angular speed of the Earth)—the slow motion case,
- $v_{max} = 2000\Omega$—the fast motion case.

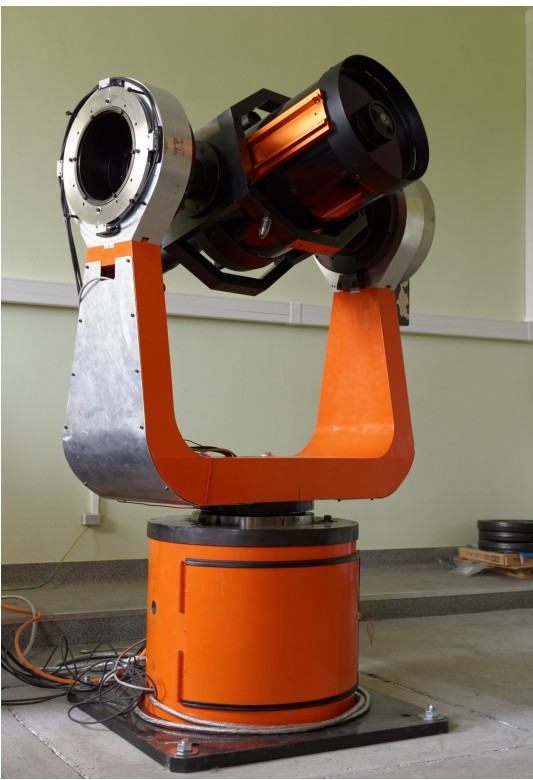

**Figure 4.** Laboratory setup equipped with the two-axis telescope mount.

Figures 5 and 6 contain time plots of the tracking errors. A significant increase in error values occurs when the velocity changes its sign due to the friction force acting upon the axis. While this temporary effect always decreases with the rise of the $\alpha$ parameter, this is not necessarily true for the tracking errors during the rest of the time, thus, the MAE criterion was calculated for every run and these are shown in Figure 7. Here, a presence of the tracking MAE minimum in Figure 7c and monotonic decrease visibly on all other plots suggest that the experimental plant with the fast desired trajectory roughly corresponds to the simulation settings with $\mu = 1$. One can assess that the system is working in a range of medium disturbances, when the CO algorithm offers the best quality for correctly estimated inertia while all other variants tend to increase their effectiveness with increase of the assumed $\hat{J}$ parameter. On the contrary, with the slow desired trajectory, the system moves into high disturbance range and the increase of assumed inertia leads to improvement of controller effectiveness in the whole considered $\alpha$ range for all controllers. This phenomenon is probably caused by the change of influence of the friction force. For the slow movement the friction is not characterized by an almost constant force and its nonzero derivative significantly changes the dynamics of the lumped disturbance. Conversely, for the fast trajectory, friction quickly reaches a constant value in a sliding regime and its influence on estimation of the dynamics is minimized. It is worth noting that the minimum in Figure 7c is obtained for $\alpha = 1.25$ instead of expected $\alpha = 1$. Whether it is caused by a character of the disturbance in the system or an incorrect choice of $\gamma$ constant remains yet unknown. Nonetheless, obtained experimental results seem to be consistent with the previously presented simulation data.

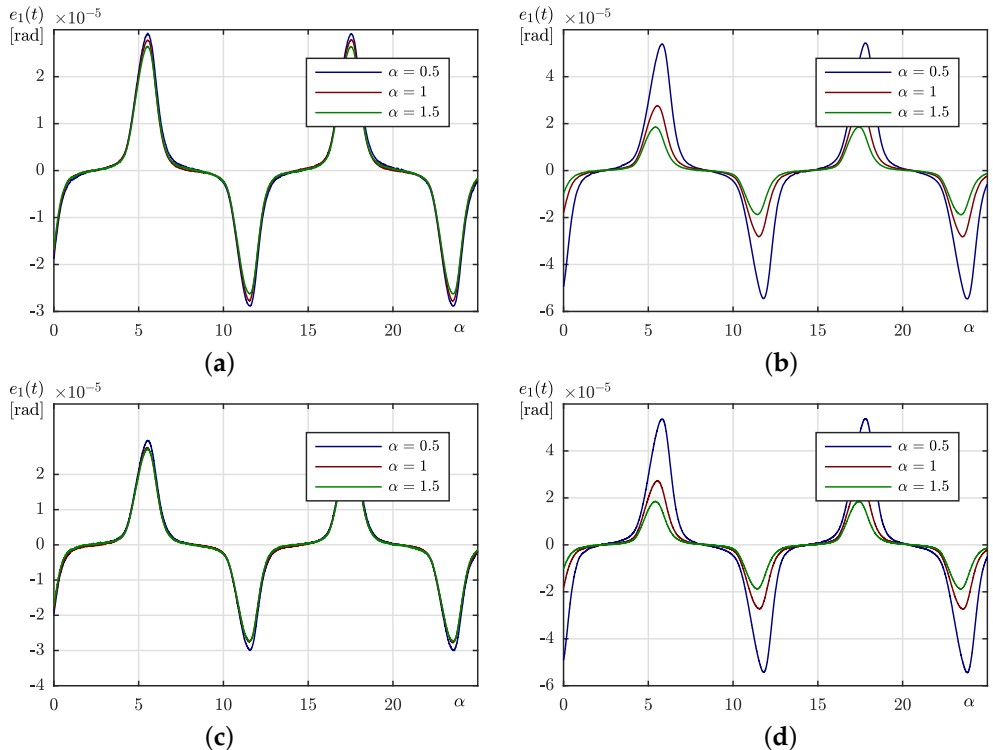

**Figure 5.** A comparison of the steady-state tracking error for slow trajectory: (**a**) CO, (**b**) CFO, (**c**) EO, (**d**) EFO.

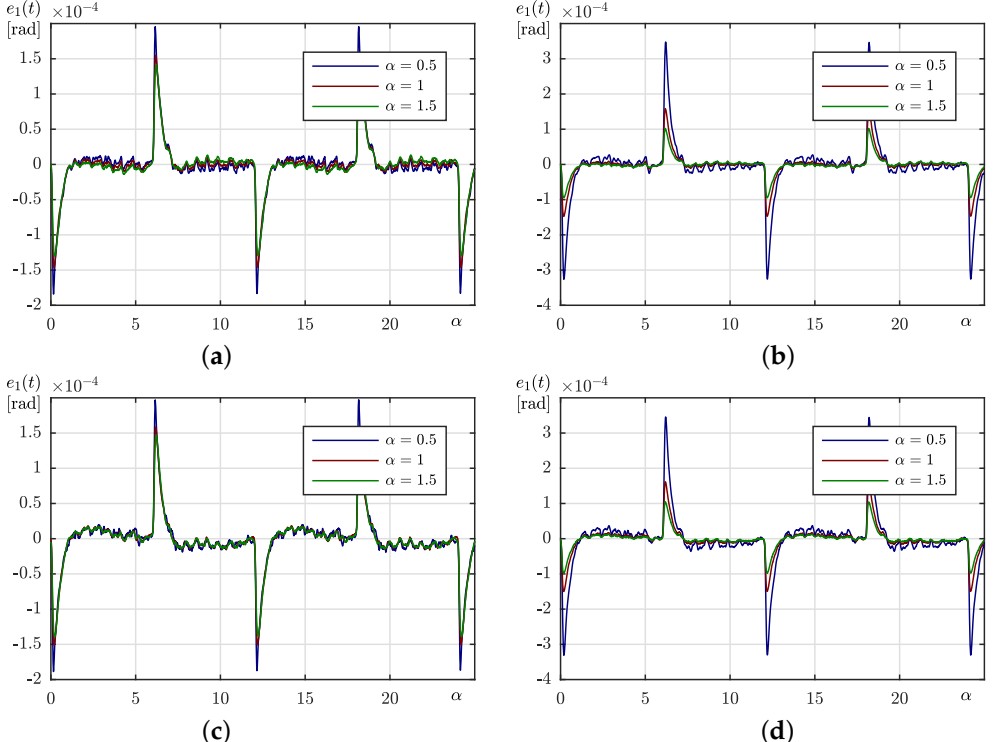

**Figure 6.** A comparison of the steady-state tracking error for fast trajectory: (**a**) CO, (**b**) CFO, (**c**) EO, (**d**) EFO.

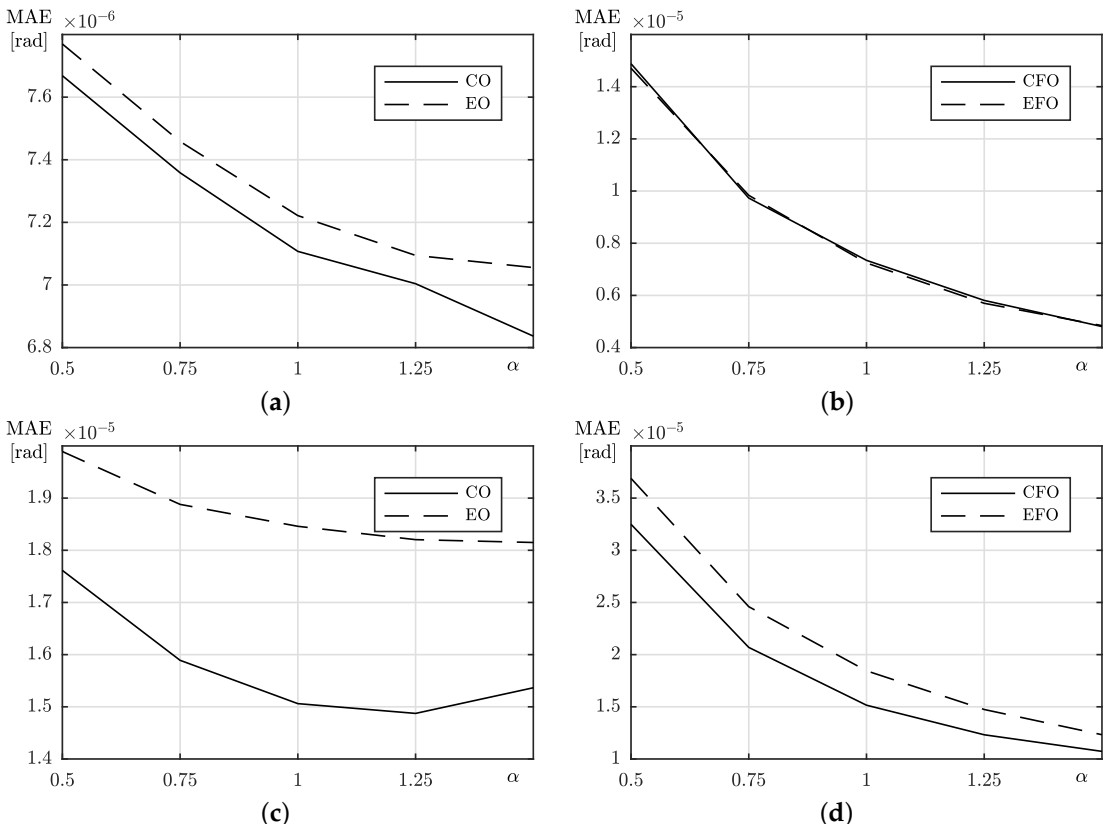

**Figure 7.** A comparison of the steady-state tracking error: (**a**,**b**) slow trajectory, (**c**,**d**) fast trajectory.

## 5. Conclusions

In this paper, a design of the distributed control for the mechanical system is described and an influence of system parameters choice on the closed-loop performance in the trajectory tracking task is investigated. Especially, the impact of chosen inertia estimate is researched. It is analytically shown that the classical ADRC and novice EADRC approaches differ by the influence of the desired trajectory $r$ and its derivatives upon the estimation dynamics only. By the means of simulations and experiments, it is revealed that, if only the observer design is taken into consideration, for a perfectly estimated inertia ADRC effectiveness depends on the character of disturbance only. Thus, an error-based observer offers decreased tracking quality due to the presence of the desired trajectory derivative inside a lumped disturbance definition even in the disturbance-free case. If value of the inertia estimate increases, this conclusion ceases to hold and EADRC keeps improving its performance, while the state based observer loses its ability to accurately estimate disturbances in the system. It is possible to conclude that if the inertia of the system is unknown to the designer, employing of the error domain active disturbance rejection controller with an overestimated inertia parameter may be a desired behavior to obtain the best tracking quality.

**Author Contributions:** Conceptualization, D.P. and R.P.; methodology, D.P. and R.P.; validation, R.P. and D.P.; formal analysis, R.P., D.P., and K.K.; resources, B.K.; writing—original draft preparation, D.P. and R.P.; writing—review and editing, D.P., B.K., R.P., and K.K.; All authors have read and agreed to the published version of the manuscript.

**Funding:** This work was partially supported by the National Science Centre (NCN) under the grant No. 2014/15/B/ST7/00429 and the Poznan University of Technology under the grant No. 33/32/SIGR/0003.

**Conflicts of Interest:** The authors declare no conflict of interest.

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
