# Peer review of "Analysis of an Impact of Inertia Parameter in Active Disturbance Rejection Control Structures"

_electronics, doi:10.3390/electronics9111801_

Round 1

Reviewer 1 Report

The chosen procedures are adequately solved the problem, the calculations are correct. The chapters are logically connected to each other. The text is suitably supplemented by pictures. However, the authenticity of the sources used cannot be verified because they are not indicated numerically, only by the symbols "?". Also in the text, the authors refer to formulas and figures that do not have numbers only "?". I consider the missing description of the x and y axes of the graph in Fig. 1 and the y-axis in the graphs in Fig. 1 to be a formal shortcoming. 3 and 4.

When describing the model by equations (1) to (4), it is a fully actuated second order mechanical system. In formula (5), the calculation is simplified and it is probably further worked with a dual-mass system. I would welcome at least a simple scheme of such a system, in which the main monitored parameters would be marked.

What is the monitored parameter α (alfa) real? Although it is given in the general form in formula (13), is it really an angle of rotation or an angular acceleration? What is its size? Also, the parameter δ is given as a derivative of a certain function (formula (14) and (19)), but what does it represent in the calculation? Because its choice of δ = 0 (chapter 2.5) greatly simplifies the calculation. It can easily be confused with the computational parameter of the solution of differential equations according to d´Alembert.

I propose to supplement the text of the article with an explanation of the mentioned parameters, correct the pictures - add a description and especially instead of "?" Add the correct numbers and only after these modifications publish the article.

Reviewer 2 Report

After the review of the article "Analysis of an impact of inertia parameter in activedisturbance rejection control structures", the reviewer states that the article may be published subject to the following changes:
1. Poor review of literature in relation to world knowledge. One gets the impression that the footnotes of the literature items are arranged so that they are full. Too many items related to a specific topic (example row 24 5 items, row 31 - 5 items)
2. Many mathematical relationships are presented, but almost the entire text does not refer to them. Such a show is pointless
3. The reviewed text contains many linguistic and grammatical errors
4. Very poor literature review. Quotations are articles that were published a long time ago (verse 169 - 44 years, verse 170 - 33 years). What are new and important these and other articles bring to the work. The work should review the current state of world knowledge, not the previous decade. The literature review is totally room for improvement.
